# PTEN/PKM2/ERα-Driven Glyoxalase 1 Overexpression Sustains PC3 Prostate Cancer Cell Growth Through MG-H1/RAGE Pathway Desensitization Leading to H_2_O_2_-Dependent KRIT1 Downregulation

**DOI:** 10.3390/antiox14091120

**Published:** 2025-09-15

**Authors:** Dominga Manfredelli, Camilla Torcoli, Marilena Pariano, Guido Bellezza, Tiziano Baroni, Vincenzo N. Talesa, Angelo Sidoni, Cinzia Antognelli

**Affiliations:** 1Division of Biosciences and Medical Embryology, Department of Medicine and Surgery, University of Perugia, 06129 Perugia, Italy; dominga.manfredelli@dottorandi.unipg.it (D.M.); camilla.torcoli@studenti.unipg.it (C.T.); marilena.pariano@unipg.it (M.P.); tiziano.baroni@unipg.it (T.B.); vincenzo.talesa@unipg.it (V.N.T.); 2Division of Anatomic Pathology and Histology, Department of Medicine and Surgery, University of Perugia, 06129 Perugia, Italy; guido.bellezza@unipg.it (G.B.); angelo.sidoni@unipg.it (A.S.)

**Keywords:** PTEN, glyoxalase 1, MG-H1, RAGE, H_2_O_2_, KRIT1, PC3, prostate cancer

## Abstract

Glyoxalase 1 (Glo1) functions as a catalyst that neutralizes methylglyoxal (MG), a highly reactive glycating agent predominantly produced during glycolysis—a metabolic pathway upregulated in cancer cells. MG primarily reacts with the amino groups of proteins (especially at arginine residues), leading to the formation of a major advanced glycation end product known as MG-derived hydroimidazolone 1 (MG-H1). We previously demonstrated in PC3 human prostate cancer (PCa) cells that the PTEN/PKM2/ERα axis promotes their aggressive phenotype by regulating the Glo1/MG-H1 pathway. In this study, after confirming our earlier findings, we investigated the downstream mechanisms of the PTEN/PKM2/ERα/Glo1/MG-H1 axis in controlling PC3 cell growth, focusing on the role of RAGE, a high-affinity receptor for MG-H1; hydrogen peroxide (H_2_O_2_); and Krev interaction trapped 1 (KRIT1), an emerging tumor suppressor. Using genetic approaches and specific inhibitors/scavengers, we demonstrated that the PTEN/PKM2/ERα/Glo1/MG-H1 axis promotes PC3 cell growth—measured by proliferation and etoposide-induced apoptosis resistance—through a mechanism involving MG-H1/RAGE pathway desensitization that leads to H_2_O_2_-mediated KRIT1 downregulation. These findings support and expand the role of PTEN signaling in PCa progression and shed light on novel mechanistic pathways driven by MG-dependent glycative stress, involving KRIT1, in this still incurable stage of the disease.

## 1. Introduction

Glyoxalase 1 (Glo1) is the main enzyme responsible for detoxifying methylglyoxal (MG), a reactive by-product of glycolysis [1,2]. MG acts as a hormetic agent, playing opposite effects according to its intracellular concentration [3,4,5]. This property is also evident in carcinogenesis [5,6], where MG can arrest [2,7,8,9,10] or promote [11,12,13] cancer growth and invasion. Through metabolizing MG, Glo1 is involved in carcinogenesis [2,11,12,14,15,16,17,18,19].

In men from Western countries, prostate cancer (PCa) represents a frequently diagnosed tumor. Surgical and radiotherapeutic interventions achieve favorable outcomes for many patients, yet a considerable fraction ultimately develop recurrent or disseminated disease with lethal consequences. Consequently, there is an urgent need for a more profound comprehension of the molecular mechanisms that enable localized/locally advanced PCas to become invasive and disseminated [20,21,22].

The role of Glo1 in PCa has long been known. In fact, Glo1 has been identified as an important contributing factor to the progression of this neoplasia [2,11,19,23], frequently through the major AGE formed by the non-enzymatic reaction of MG with arginine residues of proteins, MG-H1, and also with the involvement of the receptor for AGEs, RAGE [11].

Phosphatase and tensin homolog (PTEN) is a key tumor suppressor involved in controlling cellular proliferation and survival, mainly by modulating the PI3K/AKT/mTOR pathway [24,25,26]. It has been demonstrated that PTEN under-expression impacts PCa progression [24,27,28]. Pyruvate kinase M2 (PKM2) functions as a central glycolytic enzyme, facilitating the conversion of phosphoenolpyruvate (PEP) to pyruvate in the terminal step of glycolysis [29]. In tumor cells, PKM2 is instrumental in metabolic reprogramming, promoting the Warburg effect by favoring aerobic glycolysis over oxidative phosphorylation [30,31]. Beyond its metabolic function, PKM2 can also migrate into the nucleus, where it functions as a co-activator for various transcription factors involved in cell proliferation, survival, and tumorigenesis. Its overexpression and functional regulation have been implicated in the growth and progression of various cancers [32,33,34], including PCa [35,36].

In our previous work we demonstrated that the PTEN/PKM2/ERα signaling axis contributes to the aggressive phenotype of PC3 human PCa cells by modulating the Glo1/MG-H1 pathway. In this study, after confirming these prior observations, we explored the downstream mechanisms by which the PTEN/PKM2/ERα/Glo1/MG-H1 axis regulates PC3 cell proliferation, with particular emphasis on the involvement of hydrogen peroxide (H_2_O_2_) and Krev interaction trapped 1 (KRIT1), a tumor suppressor increasingly recognized for its role in cancer biology [37,38,39].

We found that the PTEN/PKM2/ERα/Glo1 axis promotes PC3 cell proliferation and etoposide-induced apoptosis resistance (survival) through MG-H1/RAGE pathway desensitization leading to hydrogen peroxide (H_2_O_2_)–mediated KRIT1 downregulation.

These results reinforce and broaden the understanding of PTEN signaling in PCa progression, while uncovering previously unrecognized mechanistic pathways mediated by MG-induced glycative stress and involving KRIT1, in the context of this currently incurable stage of the disease. The findings presented herein lay the groundwork for future preclinical in vivo studies aimed at evaluating the observed pathway as a potential therapeutic target to selectively slow or prevent PCa progression.

## 2. Materials and Methods

### 2.1. Materials

The cell culture medium, fetal bovine serum, penicillin/streptomycin, Laemmli buffer, LY294002 (LY) (50 µM in DMSO, 72 h), MK2206 (MK) (10 µM in DMSO, 48 h), and rapamycin (Rapam) (100 nM in DMSO, 48 h) were obtained from ThermoFisher Scientific (Milan, Italy); Roti-Block was obtained from Roth (Karlsruhe, Germany); ICI 182,780 (50 nM in DMSO, 5 h), and etoposide (20 µM in DMSO, 48 h), staurosporine (1 µM in DMSO, 48 h), MG (25 µM, 24 h), and N-acetyl-l-cysteine (NAC, 10 mM, 0.5 h) were obtained from Sigma-Aldrich (Milan, Italy). FPS-ZM1 (100 nM, 10 h) was purchased from Merck Spa (Milan, Italy). For reagents dissolved in DMSO, the final concentration of DMSO in the incubation assays was 0.01%. Control samples received an equivalent volume of the DMSO vehicle.

### 2.2. IHC of Prostate Tissues

We used formalin-fixed, paraffin-embedded (FFPE) prostate adenocarcinoma tissue samples (*n* = 60), residual material from the archives of the Division of Pathology of the University of Perugia. The authorization for this use is included in the approval by the Ethics Committee of the University of Perugia (protocol no. 2019-30). The study was conducted in accordance with the Declaration of Helsinki. Specimens were obtained from patients who underwent radical prostatectomy. Histopathological evaluation was conducted according to the Gleason score (GS) system, and tumor staging was assigned using the pTNM classification. Four-micrometer-thick FFPE tissue sections were treated with xylene to remove paraffin and subsequently rehydrated through a series of decreasing ethanol concentrations. Immunohistochemical staining was performed using the Bond III automated stainer (Leica Biosystems, Newcastle Ltd., Newcastle Upon Tyne, UK) and the Bond™ Polymer Refine Detection kit (Leica Biosystems). The sections were incubated with anti-human Glo1 antibody (1:4500 dilution) for 15 min following heat-induced epitope retrieval (HIER) at pH 6 for 20 min. Hematoxylin was used as a counterstain. The immunostained sections were assessed by two experienced pathologists (AS and GB). Staining was evaluated semi-quantitatively based on both intensity (0, 1+, 2+, 3+) and the percentage of positively stained tumor cells (0–25%, 25–50%, 50–75%, 75–100%). A composite immunoreactivity score was calculated by multiplying the intensity and percentage scores, resulting in final values of 0, 1+, 2+, 3+, 6+, or 9+ [40]. Scores of 0–2+ were considered negative or weakly positive, whereas scores of 3+, 4+, 6+, and 9+ were interpreted as indicative of moderate-to-strong expression.

### 2.3. Cell Cultures

Human prostate cancer cell lines DU145 (derived from brain metastasis) and PC3 (derived from bone metastasis) were acquired from the American Type Culture Collection (ATCC) and maintained, following the supplier’s instructions, at 37 °C in a humidified incubator with 5% CO_2_.

### 2.4. Cell Lysates

Total cellular proteins were extracted by lysing cells in RIPA buffer [19]. Nuclear protein fractions were isolated using the FractionPREP Cell Fractionation Kit (BioVision, Florence, Italy). Protein concentrations were quantified via the Lowry assay, using bovine serum albumin as the calibration standard.

### 2.5. Western Blot

SDS-PAGE and Western blot analysis were conducted following established protocols [19]. Briefly, protein samples were denatured by boiling in Laemmli buffer, separated via SDS-PAGE, and subsequently transferred onto nitrocellulose membranes. Non-specific binding sites were blocked using Roti-Block at room temperature for 1 h. The membranes were then incubated overnight at 4 °C with the appropriate primary antibodies and subsequently incubated for 1 h at room temperature with the corresponding HRP-conjugated secondary antibody. Signal detection was performed using enhanced chemiluminescence (ECL; Amersham Pharmacia, Milan, Italy).

### 2.6. Glo1, PTEN, p-AKT, and p-mTOR Evaluation

The human Glyoxalase I ELISA Kit (cod. ELH-GLYOX1-1) was used to detect Glo1 protein expression. PTEN, p-AKT, and p-mTOR protein expression was evaluated by both specific ELISA kits and Western blotting. In particular, four independent experiments were performed: three were used for ELISA analyses and the remaining one for Western blotting detection. The ELISA kits were the human PTEN ELISA Kit (cod. ab206979), the human AKT3(pS473) ELISA Kit (cod. ab270887), and the human phospho-mTOR (S2448) ELISA Kit (cod. ab279868), all from Abcam (Cambridge, UK).

### 2.7. Glo1 Enzymatic Activity Assay

Cells were collected and resuspended at 10^7^ cells/mL in 10 mM phosphate buffer (pH 7.0) supplemented with 1 mM dithiothreitol and 0.1 mM PMSF. The suspensions were homogenized using a Potter–Elvehjem homogenizer and centrifuged at 13,000× *g* for 30 min. The supernatants obtained were used for both enzymatic activity measurements and protein quantification. Glo1 activity was assessed following established protocols [19].

### 2.8. RNA Isolation, Reverse Transcription, and qRT-PCR

RNA extraction, complementary DNA synthesis, and quantitative real-time PCR (qRT-PCR) analyses were carried out as previously reported [11,19,41].

### 2.9. Measurement of H_2_O_2_ Levels

Hydrogen peroxide (H_2_O_2_) levels were measured as previously reported [42], using the Amplex^®^ Red Hydrogen Peroxide/Peroxidase Assay Kit (A22188; Invitrogen, Milan, Italy).

### 2.10. siRNA Transfection

PCa cells were transiently subjected to transfection using a mixture of ONTARGET plus SMARTpool small interfering RNA (siRNA) targeting Glo1 or PTEN and ONTARGET plus siCONTROL (siCtr) non-targeting pool as a negative control. These siRNAs were obtained from Dharmacon RNA Technologies (Carlo Erba, Milan, Italy), and the transfection process was carried out using DharmaFECT 1 transfection reagent (Dharmacon RNA Technologies), in accordance with standard protocols.

### 2.11. MG-H1 Detection

MG-H1 levels were determined using the OxiSelect™ Methylglyoxal Competitive ELISA Kit (Cell Biolabs, San Diego, CA, USA).

### 2.12. Ectopic Expression of Glo1 and PTEN

Cells were transfected with plasmids encoding Glo1 (pCMV-Glo1) or PTEN (pCMV-PTEN), as well as a control plasmid (pCMV-GFP), using transfection-ready vectors according to the manufacturer’s instructions (OriGene, Tema Ricerca, Bologna, Italy).

### 2.13. Cell Proliferation

Cell proliferation and viability were assessed using the Cell Counting Kit-8 (CCK-8, Sigma-Aldrich) following the manufacturer’s instructions.

### 2.14. Colony Formation

Colony formation was assessed as described by Ge et al. [43].

### 2.15. Apoptosis Detection

Apoptotic activity was assessed by measuring caspase-3 activation using a human active Caspase-3 ELISA Kit (Invitrogen, Milan, Italy) [12].

### 2.16. Statistical Analysis

Data were analyzed using GraphPad Prism 10.4.2 software. Comparisons between the two groups were performed with Student’s *t*-test, while multiple-group comparisons were conducted using one-way ANOVA, followed by Tukey’s post hoc test. Immunohistochemical results were evaluated using Fisher’s exact test. A *p*-value of < 0.05 was considered statistically significant.

## 3. Results

### 3.1. Glo1 Is Overexpressed in Aggressive PCa Tissues and Cell Lines

Glo1 expression was assessed by immunohistochemistry in paraffin-embedded sections from 60 patients who had undergone radical prostatectomy for either localized (T2) or locally advanced (T3) prostate cancer (PCa). Based on the Gleason score (GS), which reflects tumor aggressiveness, the patients were stratified into low-grade (LG) (*n* = 33; GS 3 + 3 (GS = 6)) and high-grade (HG) (*n* = 27; GS 4 + 4 (GS = 8, *n* = 7) and 4 + 5 or 5 + 4 (GS = 9, *n* = 20)) groups. The groups were age-matched, with a mean age of 65.5 years (median 66, range 50–73) in the LG group and 66.4 years (median 66, range 59–73) in the HG group (*p* > 0.05). All LG group patients were T2, whereas in the HG group, 70.4% of patients (*n* = 19) were T2 and 29.6% (*n* = 8) were T3. Glo1 protein expression was markedly higher in the HG group (Figure 1b) compared to the LG group (Figure 1a), confirming the significant correlation between Glo1 overexpression and PCa aggressiveness (*p* < 0.0001). Specifically, 81.8% (27/33) of LG tumors exhibited negative-to-weak staining, while 18.2% (6/33) showed moderate-to-strong expression. Accordingly, 3.7% (1/27) of HG tumors displayed negative-to-weak staining, whereas 96.3% (26/27) demonstrated moderate-to-strong expression, confirming the association of Glo1 overexpression with more aggressive disease. No significant correlation was observed between Glo1 expression and clinical stage as assessed by the TNM classification (*p* > 0.05). We also confirmed that Glo1 protein expression (Figure 1c) and enzyme-specific activity (Figure 1d) were higher in the highly aggressive PC3 cells than in the moderately aggressive DU145 cells, whose phenotypes were assessed by cell proliferation (Figure 1e) and apoptotic sensitivity (Figure 1f). Given the function of Glo1 in metabolizing MG, and consequently in preventing MG-derived hydroimidazolone 1 (MG-H1), we found decreased levels of MG-H1 in PC3 cells, which express higher levels of Glo1, compared to DU145 cells, which exhibit lower expression of this enzyme (Figure 1g).

More importantly, intracellular MG-H1 levels (Figure 2a) were modulated in accordance with Glo1 expression, in both Glo1-silenced PC3 cells (Appendix A) and Glo1-overexpressing DU145 cells (Appendix A), correlating with changes in cell proliferation (Figure 2b).

### 3.2. Glo 1 Upregulation Is Driven by the PTEN/PI3K/AKT/mTOR Pathway

PTEN functions as a tumor suppressor by modulating the PI3K/AKT/mTOR pathway, ultimately influencing different biological responses, including cell growth [44]. One prevalent alteration in PCa is PTEN deficiency, which drives overactivation of the PI3K/Akt/mTOR signaling axis [27,28,44,45]. Since Glo1 was found to be overexpressed in PC3 cells (lacking PTEN) compared to DU145 cells (PTEN wild-type), we hypothesized that this upregulation could be ascribed to the loss of PTEN. Indeed, the reintroduction of wild-type PTEN into PC3 cells (Figure 3a) resulted in a significant decrease in Glo1 expression, at both transcript and functional levels (Figure 3b), and PTEN knockdown in DU145 cells (Figure 3c) was accompanied by a substantial upregulation of Glo1 expression, at both mRNA and functional levels (Figure 3d).

As expected, the ectopic expression of PTEN in PC3 cells led to a substantial desensitization of the PI3K/AKT/mTOR pathway (Figure 4a,c). Accordingly, knockdown of PTEN in DU145 cells resulted in marked activation of the PI3K/AKT/mTOR signaling pathway (Figure 4b,d). To further investigate the involvement of this pathway in regulating Glo1 expression, PC3 cells were treated with LY294002 (LY; a selective PI3K inhibitor), MK2206 (MK; a selective AKT inhibitor), and rapamycin (Rapam; a selective mTOR inhibitor). Both transcript and functional analyses of Glo1 (Figure 4e) revealed a significant decrease in enzyme levels following inhibitor treatment.

### 3.3. Pyruvate Kinase (PK)M2 and Estrogen Receptor Alpha (ERα) Are Involved in PI3/AKT/mTOR Pathway-Triggered Glyoxalase 1 (Glo1) Upregulation

The PI3K/AKT/mTOR pathway serves as a key positive regulator of the Warburg effect. In this context, the rate-limiting glycolytic enzyme pyruvate kinase (PK) functions as a principal effector of mTOR-mediated signaling [46,47]. Tumor cells mainly express the M2 isoform of PK (PKM2) [46]. It has been established that mTOR functions as a central activator of the Warburg effect through the induction of PKM2 expression in a range of human tissues and cell lines, including PC3 cells [46]. Moreover, in malignant cells, PKM2 predominantly exists in dimeric form, a state largely resulting from phosphorylation at tyrosine 105 (PKM2 Y105). This post-translational modification has been shown to facilitate cancer cell proliferation and enhance the Warburg effect, thereby conferring a selective growth advantage to tumor cells [48]. Phosphorylation of PKM2 at tyrosine 105 (p-PKM2 Y105) in PCa has been positively associated with tumor aggressiveness and metastatic dissemination [35,49,50]. As a critical supporter of cancer cell proliferation, p-PKM2 Y105 translocates to the nucleus, where it functions as a co-activator by interacting with and facilitating the activity of various transcription factors [50]. It has been demonstrated that p-PKM2 Y105 acts as a transcriptional co-activator for ERα [51], whose expression is known to be upregulated in correlation with PCa aggressiveness [51]. We previously showed that ERα positively influences Glo1 expression. We confirmed that nuclear p-PKM2(Y105) (Figure 5a) and ERα expression (Figure 5b) in PC3 cells depends on mTOR activity, as shown using its inhibitor rapamycin. Further experiments using the ERα-degrading drug ICI 182,780 confirmed that the upregulation of Glo1 (Figure 5c) is causally linked to ERα. Overall, our data validate the signaling axis involving PI3K/AKT/mTOR, p-PKM2(Y105), and ERα.

### 3.4. Expression of the Receptor for AGEs (RAGE), Levels of Hydrogen Peroxide (H_2_O_2_), and KRIT1 Expression in DU145 and PC3 Cells

It is known that RAGE is a high-affinity receptor for MG-H1, binding it with nanomolar affinity through its V domain [52], and that the MG-H1/RAGE axis can lead to the production of H_2_O_2_ [42]. In addition, it has been described that KRIT1 is an emerging crucial player in redox homeostasis [39,53] and an increasingly recognized tumor suppressor in cancer biology [37,38,39]. Hence, we examined the expression of RAGE, the levels of H**_2_**O**_2_**, and the expression of KRIT1 in both DU145 and PC3 cell lines. Our findings indicated that all these parameters are lower in PC3 cells compared to DU145 cells (Figure 6), correlating well with the levels of MG-H1, which were also reduced in the more aggressive PC3 cells (Figure 1g), and anticorrelating with cell growth (Figure 1e,f).

### 3.5. The Desensitization of the MG-H1/RAGE/H_2_O_2_/KRIT1 Axis Sustains the Growth of Aggressive PC3 Cells

Given the aforementioned results, we hypothesized that in aggressive PC3 cells, desensitization of MG-H1/RAGE signaling could attenuate H_2_O_2_ levels, which, in turn, could downregulate KRIT1 and promote cell growth by enhancing proliferation and survival. Indeed, non-toxic 25 µM methylglyoxal (MG) (Appendix A) administered for 24 h to PC3 cells to boost MG-H1 (Figure 7a) induced RAGE expression (Figure 7b), increased H_2_O_2_ levels (Figure 7c), and promoted KRIT1 expression (Figure 7d), concurrently mitigating cell growth (Figure 7e).

Notably, upon MG treatment, inhibition of RAGE with the high-affinity RAGE-specific antagonist FPS-ZM1 [11,54,55] prevented the restoration of H_2_O_2_ levels (Figure 8a) and KRIT1 expression (Figure 8b), thereby sustaining cell growth (Figure 8c). These findings support the conclusion that desensitization of the MG-H1/RAGE axis promotes cell proliferation.

Finally, upon RAGE blockade, pre-treatment with N-acetyl-L-cysteine (NAC), a known antioxidant that increases cellular GSH levels [56], rescued KRIT1 expression (Figure 9a) and mitigated cell proliferation (Figure 9b). Altogether, these data support the conclusion that desensitization of the MG-H1/RAGE/H2O2/KRIT1 axis sustains the growth of aggressive PC3 cells.

## 4. Discussion

PCa continues to be the most frequently diagnosed cancer in men and ranks second in cancer-related deaths, with mortality largely driven by metastatic disease. Mortality in this context is predominantly attributed to metastatic progression, emphasizing the pressing need to elucidate the molecular mechanisms that drive disease advancement [57].

In this study, after confirming the role of the Glo1/MG-H1 axis driven by the PTEN/PKM2/ERα pathway in promoting the aggressive phenotype of human PCa cells, we unrevealed a downstream mechanism involving RAGE, H_2_O_2_, and KRIT1 related to the Glo1/MG-H1 axis. In particular, desensitization of the MG-H1/RAGE pathway leads to decreased H_2_O_2_ levels, which, in turn, results in KRIT1 downregulation, thereby promoting cellular growth (Figure 10).

PCa cells are known to rely on multiple compensatory signaling cascades to sustain proliferation and survival. Among these, the PI3K/Akt/mTOR cascade is a key driver of PCa aggressiveness and is tightly regulated by the tumor suppressor PTEN. PTEN is among the most commonly inactivated or mutated tumor suppressor genes in human cancers [58], with approximately 70% of advanced PCa cases exhibiting PTEN loss or consequent hyperactivation of the PI3K/Akt/mTOR pathway [59]. We here confirmed that PTEN deficiency leads to the upregulation of Glo1 through activation of the PI3K/Akt/mTOR signaling pathway and that Glo1 upregulation results in a reduction in intracellular MG-H1 levels. Activation of PI3K/AKT/mTOR signaling promotes the Warburg effect, a metabolic reprogramming often associated with cancer development and aggressiveness. In this context, pyruvate kinase (PK)—a critical enzyme controlling the final step of glycolysis—acts as a principal downstream effector of mTOR signaling [46,47]. Tumor cells predominantly express the M2 isoform of pyruvate kinase (PKM2), which is largely driven by phosphorylation at tyrosine residue 105 (PKM2(Y105)) [60]. This phosphorylated form enhances glycolytic flux [46,61], thereby contributing to the Warburg effect, which is essential for sustaining the high proliferative rate of cancer cells and conferring a selective growth advantage [62]. In PCa, elevated levels of PKM2(Y105) are positively associated with disease progression, increased aggressiveness, and metastatic potential [49,50]. Notably, phosphorylated PKM2(Y105) translocates to the nucleus, where it acts as a transcriptional co-activator by interacting with various transcription factors [63,64,65], including estrogen receptor alpha (ERα) [65]. ERα expression, in turn, has been shown to be upregulated in aggressive PCa, particularly in association with the Gleason score (GS) [66,67,68].

We previously demonstrated that ERα positively regulates the expression of glyoxalase 1 (Glo1). In this study, we confirmed that the upregulation of Glo1 is mediated by the PI3K/AKT/mTOR signaling cascade through the PI3K/AKT/mTOR/p-PKM2(Y105)/ERα axis. This pathway likely contributes to cancer cell survival during the PI3K/AKT/mTOR/p-PKM2(Y105)-driven Warburg effect by promoting a sustained glycolytic rate that supports rapid cell proliferation.

Recognizing that PI3K pathway dysregulation is a common feature of aggressive prostate cancer and diverse human cancers—including brain, breast, renal, lymphoid, cervical, and lung cancers [69]—the demonstrated pathway may represent a common feature of PI3K-driven tumorigenesis.

Advanced glycation end products (AGEs) derived from MG, including the predominant MG-H1, primarily exert their effects through the activation of RAGE, a cell surface receptor that triggers multiple intracellular signaling cascades, thereby creating a pro-oxidant environment via elevated reactive oxygen species (ROS) [70] that promote tumorigenesis [42,71]. Our findings indicate that the reduction in MG-H1, consequent to Glo1 upregulation, desensitizes the RAGE-dependent production of ROS, in particular H_2_O_2_, to reduce KRIT1 oncogenic function, culminating in cell proliferation increase and survival (etoposide-triggered apoptosis resistance). Although the MG-H1/RAGE/ROS axis is classically associated with the initiation and promotion of carcinogenesis through sustained oxidative stress and inflammatory signaling, our findings reveal a more complex and, specifically, context-dependent role in PCa. Importantly, our data uncover for the first time a functional link between this axis and the KRIT1 signaling pathway, indicating that the decrease in ROS levels downstream of MG-H1/RAGE desensitization acts as a critical modulator of intracellular signaling with potential tumor-suppressive roles. In this context, the observed downregulation of H_2_O_2_ may interfere with KRIT1-mediated signaling, leading to a loss of its regulatory functions and contributing to tumor progression. These findings add a new layer of complexity to the understanding of the MG-H1/RAGE/ROS axis in cancer biology, at least in PCa, expanding the understanding of how metabolic and signaling pathways intersect to influence PCa aggressiveness.

Moreover, the novel involvement of KRIT1 adds an additional layer of complexity to this redox-regulated network in PCa. KRIT1 has not previously been implicated in the MG-H1/RAGE/ROS axis, and its identification in this context reveals a previously unrecognized connection between glycation-driven oxidative signaling and a pathway known primarily for its roles in endothelial homeostasis, cell junction integrity, and redox balance. Our findings suggest that KRIT1 may function as a redox-sensitive signaling hub, modulating cellular responses to changes in ROS levels downstream of RAGE activation. The reduction in H_2_O_2_ caused by MG-H1 depletion appears to disrupt KRIT1-mediated signaling, potentially impairing its tumor-suppressive functions and thereby contributing to cancer progression. We speculate that this may occur through modulation of oxidative modifications of KRIT1 protein complexes or through interference with redox-sensitive transcriptional regulators, which warrants further investigation. This novel link expands the molecular framework through which oxidative stress influences PCa biology and highlights KRIT1 as a potential mediator of redox-dependent tumor control mechanisms. It also raises the possibility that KRIT1 could serve as a context-specific biomarker or therapeutic target in cancers characterized by altered glycation and redox signaling.

KRIT1 functions a scaffold protein involved in multiple signaling cascades and has a protective function against oxidative stress and inflammation. Growing evidence indicates that KRIT1 is critically involved in key redox-regulated processes, such as transcriptional signaling and autophagy, which are essential for maintaining cellular homeostasis and protection against oxidative damage. This suggests that loss of KRIT1 may exert diverse effects across various redox-sensitive pathways [72]. We speculate that KRIT1 downregulation may, in turn, modulate ROS levels, thereby establishing a self-reinforcing cycle of oxidative stress that supports proliferation and confers resistance to apoptosis. Additional plausible mechanisms could involve alterations in cytoskeletal organization or cell–cell adhesion, both of which are known to enhance cancer cell survival and aggressiveness. Future studies will be required to dissect these pathways in greater detail and to clarify the specific role of KRIT1 in PCa progression.

Interestingly, our data indicate for the first time that KRIT1 can be subjected to redox regulation, in addition to its role in controlling redox balance—an aspect that, to the best of our knowledge, has not been previously described. Therefore, our findings also add further complexity to the role of KRIT1 in redox biology.

As a key regulator of endothelial cell functions, KRIT1 is critically involved in cerebral cavernous malformation (CCM), a cerebrovascular disorder characterized by leaky, dilated vascular clusters, which can lead to seizures, neurological impairments, and intracerebral hemorrhage [39]. Despite its ubiquitous expression, limited research has explored KRIT1’s involvement in pathologies beyond CCM, including cancer [39]. In melanoma, Ercoli et al. [37] demonstrated that KRIT1 depletion enhances tumor cell proliferation, migration, and invasion; promotes β-catenin expression and nuclear translocation; and upregulates markers of inflammation and tumor plasticity. In a murine breast carcinoma model, Cici et al. [38] reported that KRIT1 protein levels are diminished or absent in metastatic tissues relative to primary tumors, suggesting a potential role in tumor progression and metastasis. In line with these previous reports [37,38], our findings reinforce the role of KRIT1 as a tumor suppressor (its reduced expression contributes to the enhanced growth of aggressive PCa cells), thereby adding a new layer of understanding to its involvement in cancer biology.

Additionally, investigating the role of KRIT1 in other cancer types and its interaction with oxidative stress pathways could provide further insights into its broader significance in tumor biology. Future research should focus on validating this mechanism in vivo and exploring potential inhibitors of the Glo1/MG-H1 axis or RAGE signaling as potential therapeutic strategies.

A limitation of this study is that, although we have demonstrated KRIT1 downregulation downstream of the PTEN/PKM2/ERα/Glo1/MG-H1 pathway and shown that exogenous MG treatment strongly supports the proposed KRIT1-associated axis, we were unable to perform KRIT1 overexpression experiments. Such experiments could have provided additional insight into the potential role of KRIT1 in regulating cell proliferation and resistance to apoptosis. Future studies will aim to investigate the functional consequences of KRIT1 restoration on cell growth and apoptotic resistance once the technical challenges of achieving sufficient overexpression are overcome. Furthermore, a deeper investigation into KRIT1 functions and its interplay with RAGE and H_2_O_2_ would offer valuable validation of the overall model, and these aspects will also be explored in future work.

Finally, some indirect evidence suggests that KRIT1 expression in human prostate tissues may be low or absent. For instance, one study reported a negative correlation between miR-21-5p expression and KRIT1 levels in vessels adjacent to the tumor, suggesting a possible downregulation of KRIT1 within the tumor microenvironment [73]. In addition, the Human Protein Atlas (www.proteinatlas.org, accessed on 1 September 2025), indeed, provides data on KRIT1 expression across various cancer types; however, these findings appear somewhat inconsistent in PCa, particularly with respect to the correspondence between mRNA and protein levels. At present, we do not have experimental data addressing KRIT1 expression in PCa tissues. This aspect will be investigated in future studies when we aim to better characterize the functional role of KRIT1 in prostate biology and cancer progression.

## 5. Conclusions

In summary, our study identified a previously unrecognized regulatory axis involving PTEN/PI3K/AKT/mTOR/p-PKM2(Y105)/ERα-driven upregulation of Glo1, which leads to MG-H1 depletion and consequent desensitization of the MG-H1/RAGE signaling cascade. This alteration reduces intracellular H_2_O_2_ levels, resulting in the dysregulation of KRIT1 and promoting PCa cell proliferation and survival. These findings significantly expand the current understanding of redox biology in PCa and position KRIT1 as a key mediator at the interface between metabolic reprogramming and redox-regulated tumor control mechanisms. Moreover, our results highlight the potential of targeting the Glo1/MG-H1/RAGE/H_2_O_2_/KRIT1 axis as a therapeutic strategy for aggressive PCa, particularly in tumors characterized by PI3K pathway hyperactivation and altered glycation signaling. Given the broader relevance of PI3K activation across multiple cancer types, KRIT1 may represent a context-specific biomarker or therapeutic target in a wide range of malignancies where glycation and redox imbalance converge to drive tumor progression. Future investigations should validate these findings in in vivo models and assess the translational potential of targeting this pathway in clinical settings. Additionally, the role of KRIT1 across different tumor types warrants further exploration, particularly in relation to its emerging function as a redox-regulated tumor suppressor.

## Figures and Tables

**Figure 1 antioxidants-14-01120-f001:**
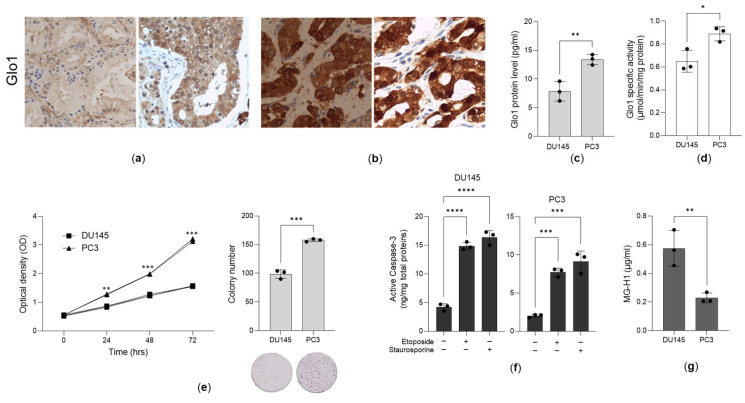
Glyoxalase 1 (Glo1) is overexpressed in aggressive prostate cancer (PCa) tissues and cell lines. Representative images of Glo1 protein expression, evaluated by immunohistochemistry, in PCa tissues from (**a**) low-grade (LG) and (**b**) high-grade (HG) patients with PCa. Magnification 200×. (**c**) Glo1 protein expression, measured by ELISA; (**d**) Glo1 specific enzyme activity, evaluated by a specific spectrophotometric method; (**e**) cell proliferation, measured by the CCK-8 assay and the clonogenic assay; (**f**) 20 µM etoposide and 1 µM staurosporine-driven apoptosis, evaluated by the increased expression of active caspase-3, using ELISA; and (**g**) MG-H1 intracellular amounts, measured by a specific ELISA kit, in moderately aggressive DU145 and highly aggressive PC3 PCa cells. The histograms indicate the mean ± SD of three different cultures. * *p* < 0.05, ** *p* < 0.01, *** *p* < 0.001, **** *p* < 0.0001.

**Figure 2 antioxidants-14-01120-f002:**
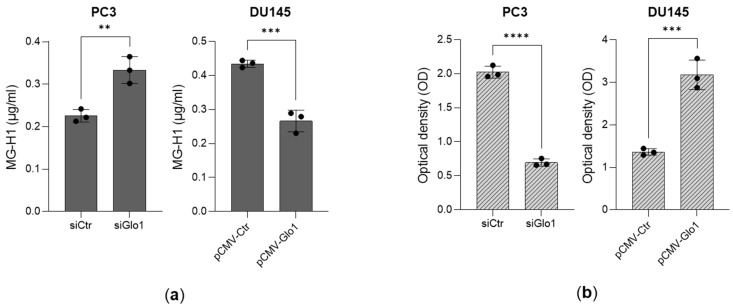
Glyoxalase 1 (Glo1) sustains prostate cancer (PCa) aggressiveness to control methylglyoxal (MG)-derived hydroimidazolone 1 (MG-H1). (**a**) MG-H1 intracellular amounts, measured by a specific ELISA kit, and (**b**) cell proliferation, measured by the CCK-8 assay, of PC3 cells upon Glo1 transient silencing (siGlo1) (**a**,**b**) and of DU145 cells upon Glo1 ectopic expression (pCMV-Glo1) (**a**,**b**). The histograms indicate the mean ± SD of three different cultures. siCtr: control (non-specific siRNA); pCMV-Ctr: control containing plasmid DNA. ** *p* < 0.01, *** *p* < 0.001, **** *p* < 0.0001.

**Figure 3 antioxidants-14-01120-f003:**
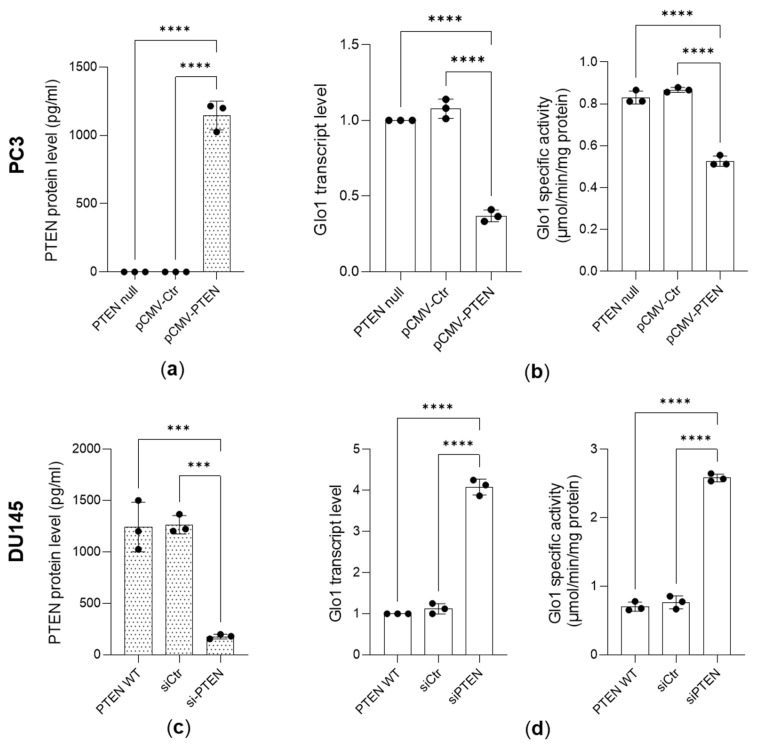
Glyoxalase 1 (Glo1) upregulation is driven by the PTEN-dependent pathway. (**a**) PTEN expression, measured by a specific ELISA kit, in PC3 cells upon PTEN ectopic expression (pCMV-PTEN) decreased (**b**) Glo1 mRNA expression and specific enzyme activity, evaluated by qRT-PCR and a specific spectrophotometric method, respectively; (**c**) PTEN expression, measured by a specific ELISA kit, in transiently PTEN-knocked-down (siPTEN) DU145 cells increased (**d**) Glo1 transcript levels and specific enzyme activity. The histograms indicate the mean ± SD of three different cultures. pCMV-Ctr: control containing plasmid DNA; siCtr: control (non-specific siRNA). *** *p* < 0.001, **** *p* < 0.0001.

**Figure 4 antioxidants-14-01120-f004:**
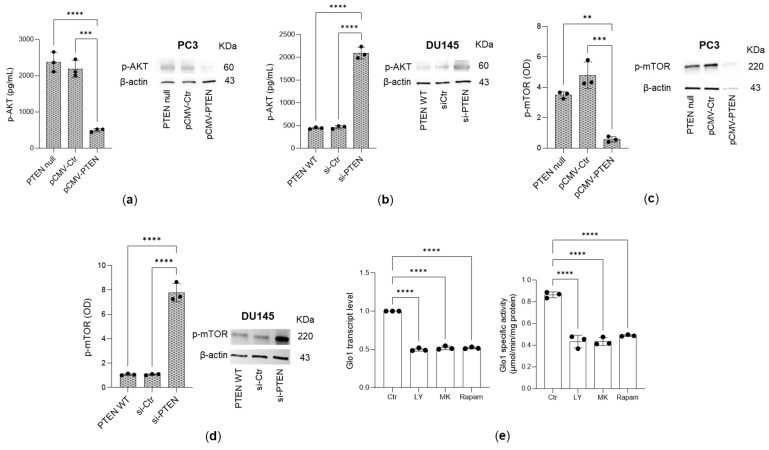
Glyoxalase 1 (Glo1) upregulation is driven by the PI3/AKT/mTOR pathway. (**a**,**c**) Ectopic expression of PTEN (pCMV-PTEN) in PC3 cells or (**b**,**d**) PTEN knockdown (siPTEN) in DU145 cells led to desensitization or activation of the PI3K/AKT/mTOR pathway, respectively. Treatment of PC3 cells with the PI3K inhibitor LY294002 (LY), the AKT inhibitor MK2206 (MK), and the mTOR inhibitor rapamycin (Rapam) further confirmed the involvement of the PI3K/AKT/mTOR signaling cascade in regulating (**e**) Glo1 at both the transcript and the functional level. The PI3/AKT/mTOR pathway was evaluated by p-AKT and p-mTOR expression measured by both specific ELISA kits and Western blotting. mRNA expression was evaluated by qRT-PCR, while specific enzyme activity was measured by a specific spectrophotometric method. β-actin served as the internal control. The histograms indicate the mean ± SD of three different cultures. pCMV-Ctr: control containing plasmid DNA; siCtr: control (non-specific siRNA). ** *p* < 0.01, *** *p* < 0.001, **** *p* < 0.0001.

**Figure 5 antioxidants-14-01120-f005:**
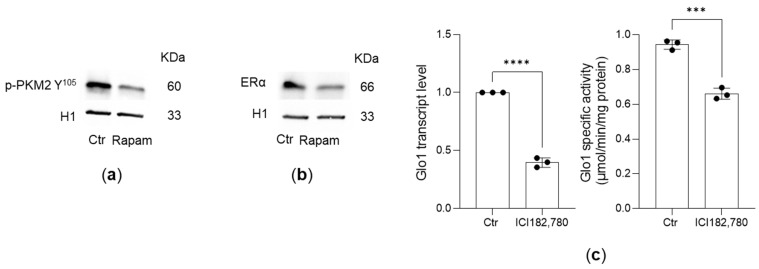
Pyruvate kinase (PK)M2 and estrogen receptor alpha (ERα) are involved in PI3/AKT/mTOR pathway-triggered glyoxalase 1 (Glo1) upregulation. (**a**) Phosphorylated PKM2 Y^105^ (p-PKM2(Y^105^)) and (**b**) ERα expression, measured by Western blotting, in nuclear extracts of PC3 cells upon rapamycin (Rapam) treatment and in controls (Ctr). (**c**) Glo1 transcript levels and specific enzyme activity upon ERα-degrading anti-estrogen ICI 182,780 treatment and in untreated (Ctr) PC3 cells. Histone H1 was used as the internal control. The histograms indicate the mean ± SD of three different cultures. *** *p* < 0.001, **** *p* < 0.0001.

**Figure 6 antioxidants-14-01120-f006:**
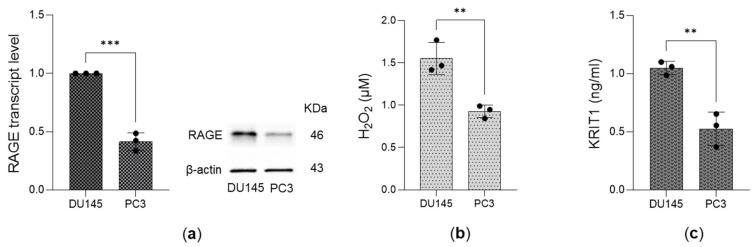
Expression of the receptor for AGEs (RAGE), levels of hydrogen peroxide (H_2_O_2_), and KRIT1 expression in DU145 and PC3 cells. (**a**) RAGE transcript levels, measured by qRT-PCR, and RAGE protein expression, evaluated by Western blotting. (**b**) H_2_O_2_ levels were measured by a specific kit, and (**c**) KRIT1 expression was studied by using a specific ELISA kit. β-actin served as the internal control. The histograms represent the mean ± SD from three independent experiments. ** *p* < 0.01, *** *p* < 0.001.

**Figure 7 antioxidants-14-01120-f007:**
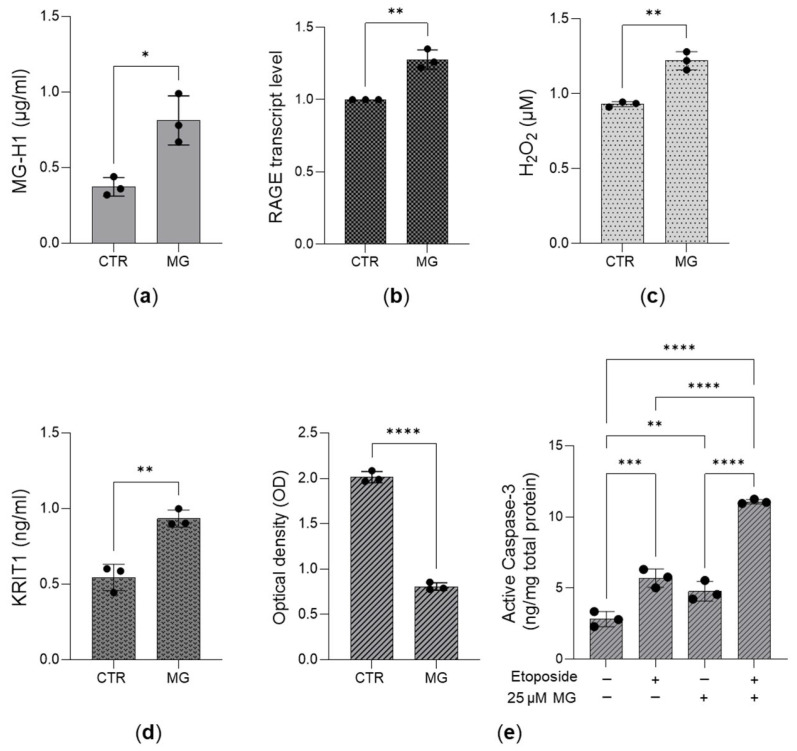
The desensitization of the MG-H1/RAGE/H2O2/KRIT1 axis sustains the growth of aggressive PC3 cells. Effect of 25 µM MG on (**a**) MG-H1 intracellular amounts, measured by a specific ELISA kit; (**b**) RAGE transcript levels, measured by qRT-PCR; (**c**) H_2_O_2_ levels, measured by a specific kit; (**d**) KRIT1 protein expression, studied by using a specific ELISA kit; and (**e**) cell growth, evaluated by both cell proliferation, as measured by the CCK-8 assay, and 20 µM etoposide-driven apoptosis, as evaluated by the increased expression of active caspase-3, using ELISA. The histograms indicate the mean ± SD of three different cultures. * *p* < 0.05, ** *p* < 0.01, *** *p* < 0.001, **** *p* < 0.0001.

**Figure 8 antioxidants-14-01120-f008:**
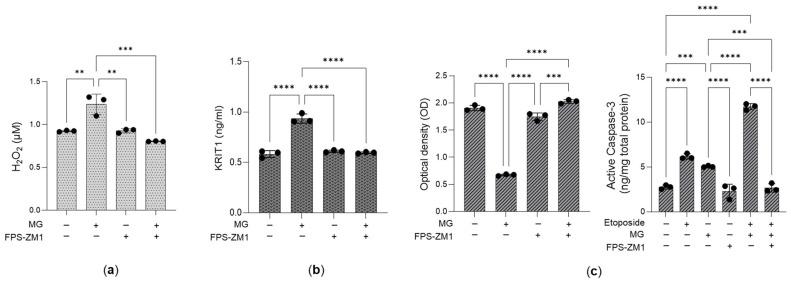
The desensitization of the MG-H1/RAGE/H2O2/KRIT1 axis sustains the growth of aggressive PC3 cells. Effect of the RAGE antagonist FPS-ZM1 (100 nM) upon 25 µM MG treatment on (**a**) H_2_O_2_ levels, measured by a specific kit; (**b**) KRIT1 protein expression, studied by using a specific ELISA kit; and (**c**) cell growth, evaluated by both cell proliferation, as measured by the CCK-8 assay, and 20 µM etoposide-driven apoptosis, as evaluated by the increased expression of active caspase-3, using ELISA. The histograms indicate the mean ± SD of three different cultures. ** *p* < 0.01, *** *p* < 0.001, **** *p* < 0.0001.

**Figure 9 antioxidants-14-01120-f009:**
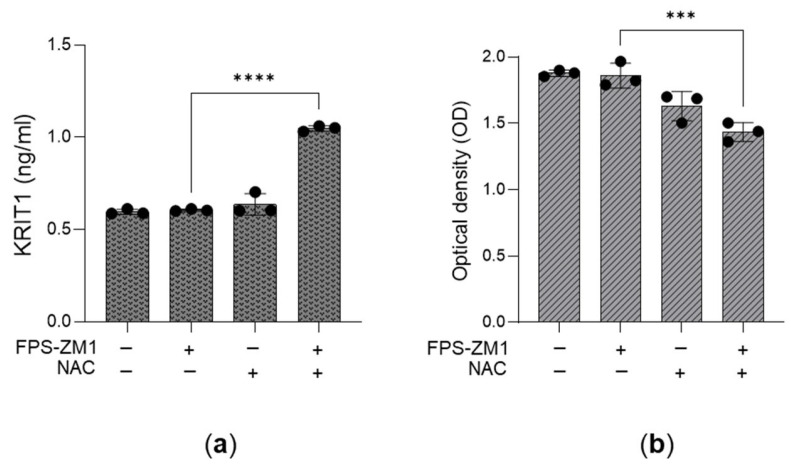
The desensitization of the MG-H1/RAGE/H2O2/KRIT1 axis sustains the growth of aggressive PC3 cells. Effect of the RAGE antagonist FPS-ZM1 (100 nM) upon NAC treatment on (**a**) KRIT1 protein expression, studied by using a specific ELISA kit, and (**b**) cell growth evaluated by cell proliferation, measured by the CCK-8 assay. The histograms indicate the mean ± SD of three different cultures. *** *p* < 0.001, **** *p* < 0.0001.

**Figure 10 antioxidants-14-01120-f010:**
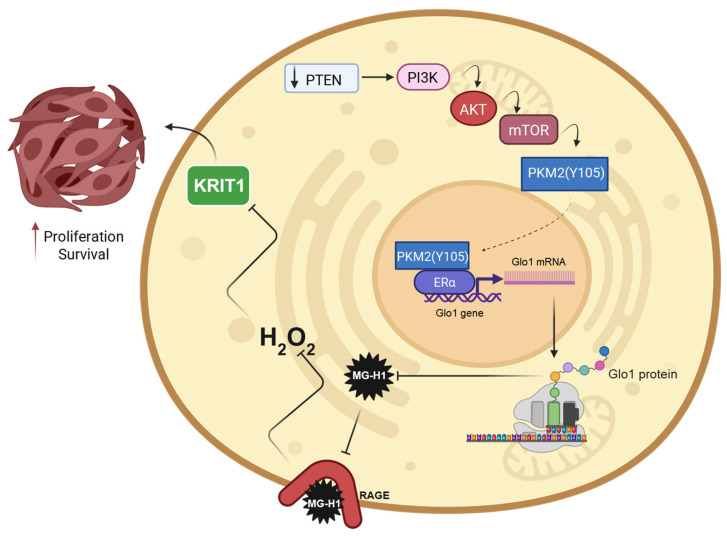
The desensitization of the MG-H1/RAGE/H2O2/KRIT1 axis sustains the growth of aggressive PC3 cells. Glo1 overexpression, driven by the PTEN/PKM2/ERα pathway, reduces intracellular MG-H1 levels, which desensitizes RAGE signaling and consequently decreases H_2_O_2_ production. This reduction leads to KRIT1 downregulation, thereby promoting cellular growth. Created with www.BioRender.com.

## Data Availability

The original contributions presented in the study are included in the article/Appendix A. Further inquiries can be directed to the corresponding author.

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
