# Peer review of "PTEN/PKM2/ERα-Driven Glyoxalase 1 Overexpression Sustains PC3 Prostate Cancer Cell Growth Through MG-H1/RAGE Pathway Desensitization Leading to H2O2-Dependent KRIT1 Downregulation"

_antioxidants, 2025, doi:10.3390/antiox14091120_

Round 1
Reviewer 1 Report
The manuscript suggests and tries to prove that PTEN/PI3K/AKT/mTOR signaling (known to be hypractive in advanced Prca), via p-PKM2(Y105) and estrogen receptor alpha ERα, drives expression of glyoxalidase (Glo1) in highly aggressive PC3 prostate cancer cells. This chain of events/signaling pathways finally depletes levels & production of advanced glycation end products measured as MG-H1 (MG-derived hydroimidazolone 1). Thats already very complicated and a bit difficult to grasp (especially when you have not previously heard of some of these genes!). Then, the lowered levels of MG-H1 somehow desensitize MG-H1/RAGE signaling,. Which, in turn (and now it gets really complicated), lowered RAGE lowers Hâ‚‚Oâ‚‚ levels, and it deregulates KRIT1 ( Krev interaction trapped 1, never heard of that one either...). Eventually, this So its very complex and involves a chain of events with at least 5 steps or levels. For purely logistic reasons alone, this required very stringent controls and measures to validate each and every one of these findings.
The authors have also used a spectrum of different techniques, which is necessary for the validation part. They used siRNA/overexpression and some specific inhibitors and pharmacologic tools (LY294002, MK2206, rapamycin, ICI-182,780, FPS-ZM1, NAC). Lat bt not least they validated expression of GLO-1 by IHC on a few patient FFPE samples which supports the association of these metabolic gene/biomarker with progression to high-grade disease.
This is quite solid study with a chain of testable and verifiable pathways that are affected, and the authors have used (as far as I can tell) the adequate controls.
There are still a few issues that remain to be fixed especially n the methodological side of Iit. If they are properly addressed, more firm and convincing conclusions can be drawn fro the studies... there are also a few errors in the materials & methods section. Plus, I have comments concerning the use of representative cell lines - this is difficult in prostate cancer. but it can be done.
I wonder why most of the experiments were done with only 1 cell line: in Figure 1, there are the "classic" but not very representative DU145 and PC3 cell lines in use. Both are highly aggressive and (in my opinion) not very representative) for prostate tissue biology or anything in which differentiation plays a role (like here). Neither PC3 nor DU145 express any of the classic PrCa pathways like androgen signaling, they don't express AR and are not even responsive to androgens and none of them shows expression of the prostate-specific differentiation markers like KLK3/PSA. So I am no t really sure how representative PC3 and DU145 for prostate biology. Its debated, to say the least. Another cell line would be really helpful, maybe one that still expresses AR and engages actively in Ar signaling (VCAP, LNCaP) and also utilizes ER signaling. Its also unclear whether any PrCa cell lines over-express GLO-1 compared to normal prostate cells? Or how is it for MG-H1 activities? Many comparable studies include "normal" or at least, non-transformed Prostate cell lines is similar studies, for this reason.
Checking proteinatlas.org for expression of GLO1 in normal vs. transformed prostate, I can see that this gene appears to be very prostate specific (Expression of GLO1 in prostate cancer - The Human Protein Atlas). However, there's not much of a difference between normal prostate cells and prostate cancer... it looks more like a tissue-specific (prostate) differentiation than a cancer marker, similar to the "classics" like KLK3. Furthermore, looking at expression levels of GLO-1 in cell lines, its apparent that DU145 and PC3 (which are not engaging in prostate-specific "acinar" differentiation) have only low levels of GLO1 and therefore I don't think they are really representative for physiological expression levels and (likely) function of GLO1 and/or MG-H1 in healthy versus cancer tissues. Cell lines VCAP and LNCAP have the highest levels, but its also not outstanding (Cell line - GLO1 - The Human Protein Atlas). Maybe the analysis of GLO1/MG-H1 in cell lines is hampered by cell culture conditions that do generally not support epithelial differentiation and maturation, which appears a condition for GLO1 expression and function. The authors should discuss this problem as a possible shortcoming of the study and the study design as a whole.
Interesting in this context is the expression of PTEN: are there any phenotypic effects of introducing PTEN or blocking PTEN expression in these 2 cell lines? I would assume PTEN is inactivated by genetic mechanisms (deletions, mutations, epigenetic) in both cell lines and introducing it could potentially have very potent effects on the viability of these cells. Vice versa, if PTN alleles are indeed knocked out in PC3 and DU145, I wonder if siRNAs should have any effects? Please clarify.
Nevertheless, the paper has logic strengths: 1) PTEN rescue/knockdown, pathway inhibitors, and ERα degradation generally support the signaling connections (regardless the questions concerning the use of cell lines above, although with ADDITIONAL cell lines the conclusions could be MUCH stronger and relevant. MG supplementation, RAGE inhibition (FPS-ZM1), and NAC probes support the involvement of Hâ‚‚Oâ‚‚/KRIT1. Technically, it appears that ll controls are useful and correctly used.
The methods list the concentrations of drugs like LY294002 were used at at 50 mM (72 h), compound MK2206 at 10 mM (48 h), and staurosporine - toxic as hell - allegedly used at 1 mM—these not likely the ones used in fact, I assume were talking about µM, not mM. Or maybe they are stock concentrations, but I assume its just a typo. Numbers are also conflicting for the staining: IHC Methods sections say n=55, Results analyze n=60 patients. Please correct to whatever are the right numbers, and update the ethics/consent text accordingly.
The use of ELISA kits is a bt unclear, I think they are all Abcam products (according to the catalog numbers) but "prodotti Gianni" is listed - I assume that's just a distributor in Italy. As far as I know, Amplex Red kit primarily detec
It would also be helpful to overexpress KRIT1 during conditions that actually promote cell growth and test whether restoring of KRIT1 expression has an effect on proliferation or apoptosis resistance of the cells... this remains unclear.
The manuscript calls 25 µM as MG “non-toxic,” yet MG also “mitigates cell growth.” Please clarify that MG at 25 µM reduces proliferation without too much or killing-level cytotoxicity, and show that there is still cell viability under these conditions. Otherwise, the logic seems quite convincing, but would be even more trustworthy if additional cell lines were used. MG supplementation increases RAGE expression, Hâ‚‚Oâ‚‚, and KRIT1, and all of that together reduces growth; FPS-ZM1 blocks these effects and sustains cell growth again; then comes NAC which rescues KRIT1 and reduces growth. These complicated up and down data are more or less consistent with the proposed model, but are a bit challenging to understand and to recapitulate. Looking deeper into the KRIT1 functions and the connection with RAGE & H2O2 could also help to validate the overall model. (I also noticed that in prostate tissues, KRIT1 is bvvery low or not even detectable... please comment on that. Has this been validated?).
Author Response
Major comments
The manuscript suggests and tries to prove that PTEN/PI3K/AKT/mTOR signaling (known to be hypractive in advanced Prca), via p-PKM2(Y105) and estrogen receptor alpha ERα, drives expression of glyoxalidase (Glo1) in highly aggressive PC3 prostate cancer cells. This chain of events/signaling pathways finally depletes levels & production of advanced glycation end products measured as MG-H1 (MG-derived hydroimidazolone 1). Thats already very complicated and a bit difficult to grasp (especially when you have not previously heard of some of these genes!). Then, the lowered levels of MG-H1 somehow desensitize MG-H1/RAGE signaling. Which, in turn (and now it gets really complicated), lowered RAGE lowers Hâ‚‚Oâ‚‚ levels, and it deregulates KRIT1 (Krev interaction trapped 1, never heard of that one either...). Eventually, this So its very complex and involves a chain of events with at least 5 steps or levels. For purely logistic reasons alone, this required very stringent controls and measures to validate each and every one of these findings. The authors have also used a spectrum of different techniques, which is necessary for the validation part. They used siRNA/overexpression and some specific inhibitors and pharmacologic tools (LY294002, MK2206, rapamycin, ICI-182,780, FPS-ZM1, NAC). Lat bt not least they validated expression of GLO-1 by IHC on a few patient FFPE samples which supports the association of these metabolic gene/biomarker with progression to high-grade disease.This is quite solid study with a chain of testable and verifiable pathways that are affected, and the authors have used (as far as I can tell) the adequate controls. There are still a few issues that remain to be fixed especially n the methodological side of Iit. If they are properly addressed, more firm and convincing conclusions can be drawn fro the studies... there are also a few errors in the materials & methods section. Plus, I have comments concerning the use of representative cell lines - this is difficult in prostate cancer. but it can be done.
Author’s response: We sincerely thank the Reviewer for the thoughtful and detailed evaluation of our manuscript. We greatly appreciate his/her careful reading and recognition of the rigor of our experimental approach and his/her positive assessment of the study’s solidity and the adequacy of controls.
Detailed comments
Comment #1: I wonder why most of the experiments were done with only 1 cell line: in Figure 1, there are the "classic" but not very representative DU145 and PC3 cell lines in use. Both are highly aggressive and (in my opinion) not very representative) for prostate tissue biology or anything in which differentiation plays a role (like here). Neither PC3 nor DU145 express any of the classic PrCa pathways like androgen signaling, they don't express AR and are not even responsive to androgens and none of them shows expression of the prostate-specific differentiation markers like KLK3/PSA. So I am no t really sure how representative PC3 and DU145 for prostate biology. Its debated, to say the least. Another cell line would be really helpful, maybe one that still expresses AR and engages actively in Ar signaling (VCAP, LNCaP) and also utilizes ER signaling. Its also unclear whether any PrCa cell lines over-express GLO-1 compared to normal prostate cells? Or how is it for MG-H1 activities? Many comparable studies include "normal" or at least, non-transformed Prostate cell lines is similar studies, for this reason. Checking proteinatlas.org for expression of GLO1 in normal vs. transformed prostate, I can see that this gene appears to be very prostate specific (Expression of GLO1 in prostate cancer - The Human Protein Atlas). However, there's not much of a difference between normal prostate cells and prostate cancer... it looks more like a tissue-specific (prostate) differentiation than a cancer marker, similar to the "classics" like KLK3. Furthermore, looking at expression levels of GLO-1 in cell lines, its apparent that DU145 and PC3 (which are not engaging in prostate-specific "acinar" differentiation) have only low levels of GLO1 and therefore I don't think they are really representative for physiological expression levels and (likely) function of GLO1 and/or MG-H1 in healthy versus cancer tissues. Cell lines VCAP and LNCAP have the highest levels, but its also not outstanding (Cell line - GLO1 - The Human Protein Atlas). Maybe the analysis of GLO1/MG-H1 in cell lines is hampered by cell culture conditions that do generally not support epithelial differentiation and maturation, which appears a condition for GLO1 expression and function. The authors should discuss this problem as a possible shortcoming of the study and the study design as a whole.
Author’s response: We thank the Reviewer for this important and thoughtful comment regarding the representativeness of PCa cell lines. We fully acknowledge that PC3 and DU145 cells, while widely used, have limitations in modeling androgen receptor (AR) signaling and prostate tissue differentiation, as they do not express AR and lack canonical markers such as KLK3/PSA.
Our study builds directly on previous work in DU145 and PC3 cells, where we demonstrated that the PTEN/PKM2/ERα axis promotes aggressive behavior via the Glo1/MG-H1 pathway. In the present study, after confirming these earlier findings, we focused on investigating the downstream mechanisms of the PTEN/PKM2/ERα/Glo1/MG-H1 axis, specifically examining the roles of RAGE, hydrogen peroxide, and KRIT1. This aspect has been clearly and explicitly highlighted from the Abstract onwards. Please, see “…We previously demonstrated in PC3 human prostate cancer (PCa) cells that the PTEN/PKM2/ERα axis promotes their aggressive phenotype by regulating Glo1/MG-H1 pathway. In the present study, after confirming our earlier findings, we investigated the downstream mechanisms of the PTEN/PKM2/ERα/Glo1/MG-H1 axis in controlling PC3 cell growth, focusing on the role of RAGE, a high-affinity receptor for MG-H1, hydrogen peroxide (Hâ‚‚Oâ‚‚) and Krev interaction trapped 1 (KRIT1), an emerging tumor suppressor.”
Our rationale for selecting, both in the previous and the present study, PC3 and DU145 cells was based on their highly aggressive, androgen-independent phenotype. These characteristics make them particularly suitable for studying advanced and treatment-resistant metastatic PCa, which represents the most clinically relevant and prognostically significant aspect of the disease. Please, see Introduction: “…In men from Western countries, prostate cancer (PCa) represents a frequently diagnosed tumor. Surgical and radiotherapeutic interventions achieve favorable outcomes for many patients, yet a considerable fraction ultimately develops recurrent or disseminated disease with lethal consequences. Consequently, there is an urgent need for a more profound comprehension of the molecular mechanisms that enable localized/locally advanced PCas to become invasive and disseminated [21–23]. The role of Glo1 in PCa has long been known. In fact, Glo1 has been identified as an important contributing factor to the progression of this neoplasia….”.
In addition, since we previously found that the Glo1/MG-H1 pathway was PTEN-dependent, we specifically selected PC3 (PTEN-null) and DU145 (PTEN-expressing) cells, both AR-negative, to avoid confounding effects from androgen receptor signaling and to allow a cleaner comparison of PTEN-dependent mechanisms in metastatic, androgen-independent PCa. It is well established that PC3 and DU145 are widely recognized and extensively used as models for aggressive castration-resistant PCa, where AR signaling is lost or bypassed. As we repeat, since our study aimed to investigate the regulation of Glo1 and MG-H1 metabolism downstream of PI3K/AKT/mTOR and ERα in a context independent of androgen signaling, these models provided an appropriate and clinically meaningful system.
Finally, PC3 cells display robust metabolic plasticity, which was essential for dissecting the signaling events linking PTEN/PI3K/AKT/mTOR, p-PKM2(Y105), ERα, and Glo1 as well as the downstream effectors. We agree that AR-positive lines such as LNCaP and VCaP could provide valuable complementary insights, particularly in linking androgen signaling to Glo1/ERα regulation. However, this was not the focus of our study. Moreover, LNCaP cells are known to have limitations related to their slower growth, increased sensitivity to culture conditions, and a narrower metabolic adaptability, which made them less practical for the broad spectrum of pharmacological and genetic manipulations required here.
As the rationale for the choice of cell models has been clearly described in the Abstract and Introduction, we believe that reiterating these points in the Discussion would be redundant. However, we remain fully available to consider any additional suggestions regarding this aspect.
Comment #2: Interesting in this context is the expression of PTEN: are there any phenotypic effects of introducing PTEN or blocking PTEN expression in these 2 cell lines? I would assume PTEN is inactivated by genetic mechanisms (deletions, mutations, epigenetic) in both cell lines and introducing it could potentially have very potent effects on the viability of these cells. Vice versa, if PTN alleles are indeed knocked out in PC3 and DU145, I wonder if siRNAs should have any effects? Please clarify.
Author’s response: We thank the Reviewer for this comment. With all due respect, it is already well known that PTEN is absent in PC3 cells (PTEN-null), whereas it is expressed in DU145 cells, which are known to harbor PTEN in a mutated or partially functional form. In our experiments, we have already demonstrated this, both in Figures 3 and 4. Moreover, we successfully reintroduced PTEN in PC3 cells and silenced it in DU145 cells, and in both cases, we observed the expected molecular and phenotypic effects, confirming the functionality of our interventions (Figures 3 and 4). Therefore, unless we have misinterpreted the question, the rationale for these observations was already clearly presented, and the Reviewer’s point had effectively been addressed in the manuscript.
Comment #3: Nevertheless, the paper has logic strengths: 1) PTEN rescue/knockdown, pathway inhibitors, and ERα degradation generally support the signaling connections (regardless the questions concerning the use of cell lines above, although with ADDITIONAL cell lines the conclusions could be MUCH stronger and relevant. MG supplementation, RAGE inhibition (FPS-ZM1), and NAC probes support the involvement of Hâ‚‚Oâ‚‚/KRIT1. Technically, it appears that ll controls are useful and correctly used.
Author’s response: We thank the Reviewer for recognizing the logical strengths of our study and we appreciate the Reviewer’s confirmation that appropriate technical controls were implemented throughout the study.
Comment #4: The methods list the concentrations of drugs like LY294002 were used at at 50 mM (72 h), compound MK2206 at 10 mM (48 h), and staurosporine - toxic as hell - allegedly used at 1 mM—these not likely the ones used in fact, I assume were talking about µM, not mM. Or maybe they are stock concentrations, but I assume its just a typo. Numbers are also conflicting for the staining: IHC Methods sections say n=55, Results analyze n=60 patients. Please correct to whatever are the right numbers, and update the ethics/consent text accordingly.
Author’s response: We thank the Reviewer for carefully pointing out these inconsistencies. We apologize for the typographical errors regarding drug concentrations: the correct concentrations are LY294002 at 50 µM (72 h), MK2206 at 10 µM (48 h), and staurosporine at 1 µM, as correctly applied in all experiments. These have now been corrected in the Methods section. Regarding the number of patients analyzed by IHC, we confirm that n = 60 is the correct number, and the Methods section has been updated accordingly. The ethics and consent statements have also been revised to reflect the accurate patient cohort. We appreciate the Reviewer’s attention to these details, which have helped improve the clarity and accuracy of our manuscript.
Comment #5: The use of ELISA kits is a bt unclear, I think they are all Abcam products (according to the catalog numbers) but "prodotti Gianni" is listed - I assume that's just a distributor in Italy. As far as I know, Amplex Red kit primarily detec
It would also be helpful to overexpress KRIT1 during conditions that actually promote cell growth and test whether restoring of KRIT1 expression has an effect on proliferation or apoptosis resistance of the cells... this remains unclear.
Author’s response: We thank the Reviewer for this comment. Regarding the ELISA kits, they are indeed Abcam products, and “Prodotti Gianni” refers simply to the distributor in Italy. We have replaced “Prodotti Gianni” with Abcam in the Methods section to clearly indicate the manufacturer.
Concerning KRIT1, we agree that overexpression experiments would provide valuable additional insight into its potential role in regulating cell proliferation and apoptosis resistance. Indeed, we had attempted to perform KRIT1 overexpression experiments alongside the rest of the experimental work, but, unfortunately, we encountered technical difficulties in achieving sufficient expression levels for robust functional analysis. Since the work was already comprehensive/complex and revealed a novel and interesting mechanism downstream of the PTEN pathway, whose plausibility was strengthened by the experiments where we exposed cells to methylglyoxal and FPS-ZM1 or NAC (Figures 7-9) [treatment with methylglyoxal appears to induce coordinated responses among the downstream molecules implicated in the proposed mechanism (RAGE, Hâ‚‚Oâ‚‚, and KRIT1)],we did not pursue it further at this stage, but we plan to address it in future research.
We have now included a statement in the Discussion (please, see the last paragraph) acknowledging this limitation and highlighting that future studies will aim to investigate the functional consequences of KRIT1 restoration on cell growth and apoptotic resistance, once technical issues are resolved. Regarding the Reviewer’s comment on Amplex Red kit, the sentence appears to be incomplete, and we would appreciate clarification in order to fully understand the point being raised.
Comment #6: The manuscript calls 25 µM as MG “non-toxic,” yet MG also “mitigates cell growth.” Please clarify that MG at 25 µM reduces proliferation without too much or killing-level cytotoxicity, and show that there is still cell viability under these conditions. Otherwise, the logic seems quite convincing, but would be even more trustworthy if additional cell lines were used. MG supplementation increases RAGE expression, Hâ‚‚Oâ‚‚, and KRIT1, and all of that together reduces growth; FPS-ZM1 blocks these effects and sustains cell growth again; then comes NAC which rescues KRIT1 and reduces growth. These complicated up and down data are more or less consistent with the proposed model, but are a bit challenging to understand and to recapitulate. Looking deeper into the KRIT1 functions and the connection with RAGE & H2O2 could also help to validate the overall model. (I also noticed that in prostate tissues, KRIT1 is bvvery low or not even detectable... please comment on that. Has this been validated?).
Author’s response: We thank the Reviewer for this detailed and insightful comment. Regarding the use of 25 µM MG, we agree that it is important to clarify that MG reduces cell proliferation without inducing substantial or lethal cytotoxicity. Preliminary MTT assays were performed to assess cell viability, and these data have now been included in the Supplementary Materials, confirming that a significant fraction of cells remains viable at 25 µM MG. This supports our designation of this concentration as “non-toxic” for the purposes of the experiments. Regarding the Reviewer’s suggestion to include additional cell lines, this point has already been addressed and justified above (Comment#1).
We fully agree that a deeper investigation into KRIT1 functions and its connection with RAGE and H2O2 would provide valuable additional validation of the overall model. However, such studies are beyond the scope of the present work, which is already complex and focused on dissecting the downstream mechanisms of the PTEN/PKM2/ERα/Glo1/MG-H1 axis, as highlighted in the manuscript. We have now included a statement in the Discussion noting this as a limitation and emphasizing that future studies will aim to explore these aspects. Please, see the last paragraph of the Discussion.
Finally, we thank the Reviewer for the last important observation (“I also noticed that in prostate tissues, KRIT1 is bvvery low or not even detectable... please comment on that. Has this been validated?”). As far as we know, currently, no published studies have directly examined KRIT1 expression in human prostate tissues. However, there is some indirect evidence indicating low or absent KRIT1 expression in this context. For example, one study reported a negative correlation between miR-21-5p expression and KRIT1 in vessels adjacent to the tumor, suggesting a possible downregulation of KRIT1 in the tumor microenvironment [PMID: 34088891]. Moreover, the Human Protein Atlas indeed provides data on KRIT1 expression across various cancer types; however, these findings appear somewhat inconsistent in PCa, particularly with respect to the correspondence between mRNA and protein levels. At the currents stage, we do not have experimental data directly addressing KRIT1 expression in PCa tissues as suggested by the Reviewer. This aspect will be investigated in future studies when we aim to better characterize the functional role of KRIT1 in prostate biology and cancer progression. We have now included a statement in the Discussion highlighting this point and noting it as a limitation of the present work (please, see the last paragraph of Discussion).
We hope these revisions strengthen the overall conclusions of the study. We are grateful for your constructive suggestions and the opportunity to improve our manuscript.

Reviewer 2 Report
The article by Manfredelli et al. investigates Hâ‚‚Oâ‚‚-dependent KRIT1 deregulation caused by glyoxalase 1 overexpression in PC3 prostate cancer cells. This study contributes to the understanding of PTEN-dependent signaling pathways and their implications for metabolic regulation and glycobiology in cancer cells. The presented results are promising and merit further scientific recognition, provided that the Authors address several points to improve the overall quality of the manuscript.
- The manuscript includes data from clinical samples but focuses mainly on tumor grading and staging. If available, incorporating survival or disease progression data would significantly enhance the translational relevance of the study.
- The discussion on the influence of KRIT1 on prostate cancer cell proliferation lacks detail regarding the specific molecular pathways involved. Please expand this section to better explain KRIT1’s role.
- To validate the proposed role of KRIT1, additional mechanistic studies involving its knockdown and overexpression are strongly recommended.
- The mechanistic link between Hâ‚‚Oâ‚‚ levels and KRIT1 deregulation remains insufficiently explained. Please clarify how altered Hâ‚‚Oâ‚‚ levels influence KRIT1 function.
- Please provide higher-resolution immunohistochemistry images.
Author Response
Major comments
The article by Manfredelli et al. investigates H2O2-dependent KRIT1 deregulation caused by glyoxalase 1 overexpression in PC3 prostate cancer cells. This study contributes to the understanding of PTEN-dependent signaling pathways and their implications for metabolic regulation and glycobiology in cancer cells. The presented results are promising and merit further scientific recognition, provided that the Authors address several points to improve the overall quality of the manuscript.
Author’s response: We sincerely thank the Reviewer for his/her positive assessment of our work and for recognizing the relevance of our findings in the context of PTEN-dependent signaling, metabolic regulation, and glycobiology in PCa. We also appreciate the constructive suggestions provided, which we believe will further strengthen the clarity and overall quality of our manuscript.
Detailed comments
Comment #1: The manuscript includes data from clinical samples but focuses mainly on tumor grading and staging. If available, incorporating survival or disease progression data would significantly enhance the translational relevance of the study.
Author’s response: We thank the Reviewer for this valuable suggestion. Unfortunately, survival and disease progression data were not available for the patient cohort included in our study, which was primarily characterized in terms of tumor grading and staging. We fully agree that incorporating such information would greatly enhance the translational relevance of the findings, and we will certainly take this into consideration in the design of future studies, where prospective collection of clinical follow-up data will allow us to strengthen the clinical impact of our observations.
Comment #2: The discussion on the influence of KRIT1 on prostate cancer cell proliferation lacks detail regarding the specific molecular pathways involved. Please expand this section to better explain KRIT1’s role.
Author’s response: We thank the Reviewer for his/her constructive suggestion. We have now expanded the Discussion to provide additional context on KRIT1’s potential mechanisms of action. In particular, we speculate that KRIT1 deregulation may in turn modulate ROS levels, thereby establishing a self-reinforcing cycle of oxidative stress that supports proliferation and confers resistance to apoptosis. Additional plausible mechanisms could involve alterations in cytoskeletal organization or cell–cell adhesion, both of which are known to enhance cancer cell survival and aggressiveness. Future studies will be required to dissect these pathways in greater detail and to clarify the specific role of KRIT1 in PCa progression. Please, see Discussion, lines 692-697.
Comment #3: To validate the proposed role of KRIT1, additional mechanistic studies involving its knockdown and overexpression are strongly recommended.
Author’s response: We thank the Reviewer for this comment. We agree that overexpression experiments would provide valuable additional insight into its potential role in regulating cell proliferation and apoptosis resistance. Indeed, we had attempted to perform KRIT1 overexpression experiments, but, unfortunately, we encountered technical difficulties in achieving sufficient expression levels for robust functional analysis. Since the work was already comprehensive/complex and revealed a novel and interesting mechanism downstream of the PTEN pathway, whose plausibility was strengthened by the experiments where we exposed cells to methylglyoxal and FPS-ZM1 or NAC (Figures 7-9) [treatment with methylglyoxal appears to induce coordinated responses among the downstream molecules implicated in the proposed mechanism (RAGE, Hâ‚‚Oâ‚‚, and KRIT1)],we did not pursue it further at this stage, but we plan to address it in future research. We have now included a statement in the Discussion (please, see the last paragraph) acknowledging this limitation and highlighting that future studies will aim to investigate the functional consequences of KRIT1 restoration on cell growth and apoptotic resistance, once technical issues are resolved.
Comment #4: The mechanistic link between H2O2 levels and KRIT1 deregulation remains insufficiently explained. Please clarify how altered H2O2 levels influence KRIT1 function.
Author’s response: We thank the Reviewer for highlighting this point. We have revised the manuscript to include the speculation that this may occur through modulation of oxidative modifications of KRIT1 protein complexes or via interference with redox-sensitive transcriptional regulators, which warrants further investigation. Please see Discussion, lines 657–659.
Comment #5: Please provide higher-resolution immunohistochemistry images.
Author’s response: We thank the Reviewer for this suggestion. Upon manuscript submission, we are asked to upload high-resolution immunohistochemistry images (the PowerPoint file originally provided to the journal contains images at higher resolution). The apparent loss of resolution occurs when these images are copied from the PowerPoint slides into the Word manuscript submitted for review, which is a technical limitation of the file conversion process. We hope that the figures in the published version will reflect the original submitted PowerPoint high-resolution images.
We hope these revisions strengthen the overall conclusions of the study. We are grateful for your constructive suggestions and the opportunity to improve our manuscript.

Round 2
Reviewer 1 Report
Luckily, the authors fixed most or all of the reviewers' issues with the manuscript; for example, they fixed the concentration numbers for LY/MK/staurosporine in Methods, and captions already had the correct final doses. The number of IHC stainings (60) was fixed throughout the text. Vendor names were missing, but have been added. The questions about non-toxic concentrations were fixed (e.g., in Fig. S3).
What has NOT been done is adding more cell lines (e.g., for AR expression), and they didn't do the KRIT1 rescue experiments that were suggested. That's probably acceptable for a purely mechanistic paper, and this is not "Cell," so the relegation of such experiments into the distant future can be tolerated/accepted. Instead, the authors acknowledged not doing these suggested additions as a limitation for the plausibility of their main findings: "MG-H1/RAGE pathway desensitization leading to H2O2-dependent KRIT1 deregulation" (= title). It may be so, or maybe it's not. We don't know 100% and we wont find out with this publication. The unaddressed experimental tasks are now explicitly acknowledged as future work, which is reasonable here and acceptable, I do not think any additional minor reviewing rounds would significantly change anything about this - therefore the paper should be accepted but asking the authors to do all the quality control outlined below.
there are still a number of rather small issues, I do not think they need another round of (minor) modifications, but they can be addressed by the authors nevertheless in "production":
1) there is still one type related to the drug concentrations: etoposide 150 mM in DMSO, 48 h in section 2.1 of "Materials" (line 87). That cannot be correct, right? What stock solution was used? Just check. There may be more of these, the authors need to do careful proofreading of the text.
2) occasionally, figures arent cited correctly, such as "Figure e" when "Figure 7e" is meant (line 322).
3) there are still english language typos; again I wont claim to have found all of them, but they persist: For example, "sustain the cell growth" should read "sustains the cell growth". MOst of them are minor. But they should be in the title of a section, like they are here: "3.5. The desensitization of the axis MG-H1/RAGE/H2O2/KRIT1 sustain the cell growth of aggressive PC3 cells" (line 314). This typo occurs 4 times in the text. Also it should read "administered" and not "administrated" - sounds similar, but wrong. [Why dont you simply use Grammarly or another free proofreading/grammar tool). Also, capitalization isnt consistent: sometimes its "β-actin" and other times its "β-Actin", just be consistent.
4) KRIT1 "deregulation", what exactly does that mean? Up or down? i dont think its clear
Author Response
Major comments: Luckily, the authors fixed most or all of the reviewers' issues with the manuscript; for example, they fixed the concentration numbers for LY/MK/staurosporine in Methods, and captions already had the correct final doses. The number of IHC stainings (60) was fixed throughout the text. Vendor names were missing, but have been added. The questions about non-toxic concentrations were fixed (e.g., in Fig. S3). What has NOT been done is adding more cell lines (e.g., for AR expression), and they didn't do the KRIT1 rescue experiments that were suggested. That's probably acceptable for a purely mechanistic paper, and this is not "Cell," so the relegation of such experiments into the distant future can be tolerated/accepted. Instead, the authors acknowledged not doing these suggested additions as a limitation for the plausibility of their main findings: "MG-H1/RAGE pathway desensitization leading to H2O2-dependent KRIT1 deregulation" (= title). It may be so, or maybe it's not. We don't know 100% and we wont find out with this publication. The unaddressed experimental tasks are now explicitly acknowledged as future work, which is reasonable here and acceptable, I do not think any additional minor reviewing rounds would significantly change anything about this - therefore the paper should be accepted but asking the authors to do all the quality control outlined below.
Author’s response: We sincerely thank the Reviewer for the constructive comments, which have helped us to improve our study.
Detailed comments
there are still a number of rather small issues, I do not think they need another round of (minor) modifications, but they can be addressed by the authors nevertheless in "production":
Comment #1
there is still one type related to the drug concentrations: etoposide 150 mM in DMSO, 48 h in section 2.1 of "Materials" (line 87). That cannot be correct, right? What stock solution was used? Just check. There may be more of these, the authors need to do careful proofreading of the text.
Author’s response: We thank the Reviewer for pointing out this inconsistency. The etoposide concentration was reported correctly in the corresponding figure legends but not in the Materials and Methods section. We have now corrected the error to ensure consistency throughout the manuscript, changing 150 mM to 20µM.
Comment #2
occasionally, figures arent cited correctly, such as "Figure e" when "Figure 7e" is meant (line 322).
Author’s response: We thank the Reviewer for noticing this mistake. The incorrect figure citation has now been corrected to ‘Figure 7e’ in the revised manuscript.
Comment #3
there are still english language typos; again I wont claim to have found all of them, but they persist: For example, "sustain the cell growth" should read "sustains the cell growth". MOst of them are minor. But they should be in the title of a section, like they are here: "3.5. The desensitization of the axis MG-H1/RAGE/H2O2/KRIT1 sustain the cell growth of aggressive PC3 cells" (line 314). This typo occurs 4 times in the text. Also it should read "administered" and not "administrated" - sounds similar, but wrong. [Why dont you simply use Grammarly or another free proofreading/grammar tool). Also, capitalization isnt consistent: sometimes its "β-actin" and other times its "β-Actin", just be consistent.
Author’s response: We thank the Reviewer for carefully pointing out these typographical and grammatical inconsistencies. We have corrected the identified errors. The manuscript has been thoroughly proofread to minimize any remaining issues.
Comment #4
KRIT1 "deregulation", what exactly does that mean? Up or down? i dont think its clear
Author’s response: We thank the Reviewer for the comment To avoid ambiguity, we have replaced the term ‘deregulation’ with a more precise wording to indicate that KRIT1 is negatively regulated (“downregulation”). The text has been revised accordingly.
We are grateful for your constructive suggestions and the opportunity to improve our manuscript.
Reviewer 2 Report
The Authors have addressed all my comments and substantially improved the quality of the manuscript.
I have no further inquiries.
Author Response
Comment: The Authors have addressed all my comments and substantially improved the quality of the manuscript.
Authors’ response: We sincerely thank the reviewer for his/her careful reading and constructive comments. We are glad that our revisions have addressed his/her concerns and improved the manuscript.